# Natural Inhibitors of P-glycoprotein in Acute Myeloid Leukemia

**DOI:** 10.3390/ijms24044140

**Published:** 2023-02-18

**Authors:** Manuela Labbozzetta, Paola Poma, Monica Notarbartolo

**Affiliations:** Department of Biological, Chemical and Pharmaceutical Science and Technology (STEBICEF), University of Palermo, 90128 Palermo, Italy

**Keywords:** acute myeloid leukemia, multidrug resistance, P-glycoprotein, natural substances

## Abstract

Acute myeloid leukemia (AML) remains an insidious neoplasm due to the percentage of patients who develop resistance to both classic chemotherapy and emerging drugs. Multidrug resistance (MDR) is a complex process determined by multiple mechanisms, and it is often caused by the overexpression of efflux pumps, the most important of which is P-glycoprotein (P-gp). This mini-review aims to examine the advantages of using natural substances as P-gp inhibitors, focusing on four molecules: phytol, curcumin, lupeol, and heptacosane, and their mechanism of action in AML.

## 1. Introduction

The new therapeutic strategies available today for cancer treatment include targeted therapies, which have significantly increased patient survival [1]. Despite the current and emerging therapeutic landscape for the treatment of cancer and an improved understanding of mechanisms of resistance to conventional therapies [2], primary and secondary drug resistance remains a substantial problem for most patients. Acquired resistance is often determined by the overexpression of efflux pumps and develops rapidly. Acute myeloid leukemia is no exception in this sense because a subgroup of patients often develops resistance and becomes refractory to all available therapies. Among the efflux pumps, P-glycoprotein (P-gp) is certainly the most studied as a pharmacological target. However, the inhibitors tested so far have shown toxicity incompatible with use in the clinic [3,4,5]; therefore, especially recently, the attention has shifted towards natural molecules, characterized by low toxicity. These substances can act specifically on the overexpression of the pump, and while they do not show toxicity on normal cells, their effects on the physiological function of the P-gp are unknown. In this case, targeting a specific action on the P-gp, expressed by the neoplastic cell, could represent a good strategy.

In particular, we present the mechanism of action of four natural substances belonging to different categories: terpenoids, polyphenols, and hydrocarbons. Phytol is acyclic hydrogenated diterpene alcohol, lupeol is a pentacyclic triterpenoid, curcumin is a polyphenol and heptacosane is a straight-chain alkane with 27 carbon atoms. All these compounds are capable of targeting P-gp by inhibiting its expression at the transcriptional level and/or by acting as substrate inhibitors of the same protein, yet they do so through different mechanisms.

## 2. Acute Myeloid Leukemia

Acute myeloid leukemia is the most common acute form of leukemia, the onset of which is related to age, with an incidence of 3–4 cases per 100,000 people a year. It is a heterogeneous neoplastic disease, characterized by the acquisition of a broad spectrum of molecular alterations, with a higher frequency of occurrence in the elderly population (60–65 years) [6]. However, a good percentage also occurs in pediatric ages and in young people (15–20% of leukemias occur in children and approximately 33% in adolescents and young adults), with a 5-year overall survival (OS) rate over the last 20 years of 60–75% in children and only 50–60% in young adults (15–39 years) [7]. Regarding the therapeutic possibilities, the standard “7 + 3” chemotherapy (continuous infusion of cytarabine over 7 days with the addition of an anthracycline, given daily for the first 3 days) often has a weak outcome, with only about 40% complete remission rate and a median OS of 12–18 months. Contextually, knowledge of the great molecular heterogeneity and genetic diversity of AML has led to the discovery of several new promising therapies for the treatment of this disease. Recently, new therapeutic agents have been approved by the FDA: CPX 351, a liposomal formulation of fixed-dose cytarabine and daunorubicin; venetoclax, an oral highly selective inhibitor of the anti-apoptotic protein, BCL-2, in combination with a hypomethylating agent, decitabine or azacitidine; FLT3 (Fms-like tyrosine kinase 3) inhibitors, midostaurin, and gilteritinib, either alone or in combination with conventional chemotherapies; IDH1/2 (isocitrate dehydrogenase 1 and 2) inhibitors, ivosidenib, and enasidenib; the conjugated anti-CD33-antibody, gemtuzumab ozogamicin; glasdegib, a selective inhibitor of hedgehog signaling [8,9,10]. These new targeted therapies, compared to standard chemotherapy, have improved response rates and outcomes for selected AML patients in different clinical scenarios. Furthermore, sorafenib, a multi-kinase inhibitor of RAF, KIT, and FLT3 kinases could represent a useful drug in AML. Recently, it has been evaluated in a phase II trial (SORMAIN) as a maintenance therapy and in an open-label randomized phase III trial in adult patients with FLT3-ITD (internal duplications in tandem) mutated AML, who had already undergone allogeneic hematopoietic stem cell transplantations. Both studies showed a good survival benefit for AML patients [11,12].

Many types of cancer express ABC transporters, such as colorectal cancer, liver cancer, acute myeloid leukemia (AML), and breast cancer. In approximately 35% of AML patients, the overexpression of P-gp causes the failure of anticancer therapies. Indeed, P-g expression has been detected in approximately one-third of patients with AML at diagnosis and in more than half of the patients in remission. Furthermore, higher levels have also been observed in certain subtypes, including secondary leukemia. In AML, P-gp expression is associated with a lower rate of complete remission, disease-free survival, and overall survival [13,14,15].

Unfortunately, even with the most innovative therapies, we are still observing the development of drug resistance, which remains one of the key events affecting therapeutic failure. In fact, there are refractory or resistant forms, in which the targeted therapies can be made ineffective by genetic mutations. Multidrug resistance (MDR) concerns most AML patients, which can develop promptly, and in these cases, allogeneic stem cell transplantation becomes the only choice. The research of molecules able to overcome this condition is therefore urgent.

## 3. The Multidrug Resistance Mediated by Efflux Pumps

The phenomenon of multidrug resistance is complex and can spring from different mechanisms, such as the overexpression of members of the adenosine triphosphate binding cassette (ABC) transporters, which includes P-gp, a multidrug resistance-1 protein-coding gene (MDR1 or ABCB1), the multidrug resistance associated protein (MRP1 or ABCC1), and the breast cancer resistance protein (BCRP or ABCG2). These transporters are implied in the genesis of the MDR phenotype in various tumor models [13].

To date, the most studied strategy to limit MDR is by blocking the efflux of the chemotherapy drug through the inhibition and modulation of the efflux pump functions, by either chemosensitizing, revertant inhibitors of MDR, or modulators of P-gp. As mentioned, the main pharmacological strategy of preclinical and clinical research, to overcome MDR and revert the resistant phenotype, is represented by the co-administration of an anticancer chemotherapy drug and P-gp substrate-inhibitor compounds, to restore the effective therapeutic concentration of the drug in the tumor-resistant cells. Among the many compounds tested, a classification was made over time between first-, second-, and third-generation inhibitors. Moreover, these compounds have different mechanisms of action since some act as competitive antagonist substrates, others as modulators, and some as pure inhibitors. The first-generation modulators include very different molecules as well as calcium channel blockers (e.g., verapamil), calmodulin antagonists (trifluoperazine), antiarrhythmic agents, and immunomodulators (e.g., quinidine and cyclosporine A). The second-generation modulators are characterized by greater potency and efficacy, yet remain very toxic; they are often structural derivatives of first-generation compounds such as dexverapamil, the R-enantiomer of verapamil. The third-generation modulators are derived from quantitative structure–activity studies and combinatorial chemistry techniques. These compounds (e.g., laniquidar, ONT-093, zosuquidar, and tariquidar) cause fewer pharmacokinetic and pharmacodynamic interactions because they only minimally interfere with the cytochrome P-450 system. In addition, due to their high affinity for P-gp, they are used at very low concentrations and are, therefore, not very toxic [16]. It is important to consider that P-gp is normally present in healthy cells because it performs many physiological functions, including providing protection against lipophilic and cytotoxic xenobiotic molecules. Inhibition of the physiological function of P-gp increases the danger of intoxication due to increased drug absorption and the reduced excretion of metabolites. However, despite all the progress that has been made in designing strategies to overcome MDR in humans, none of the currently known P-gp inhibitors can be considered an ideal inhibitor, capable of safely and completely reversing MDR in cancer cells [17]. Moreover, none have ever made it to the clinic due to multiple factors, including numerous drug interactions, dose-limiting toxicity, and a lack of specificity for P-gp [3,6,18]. In the case of acquired drug resistance, P-gp is typically expressed in response to chemotherapy and, thus, is induced by the administration of anticancer chemotherapy drugs; however, non-toxic ABC transporter modulators possessing synergistic activity with conventional chemotherapy drugs can be used prophylactically to prevent the incidence of the MDR phenotype.

## 4. Natural Inhibitors of P-glycoprotein in Leukemia

Many researchers are focusing on products of natural origin that have been shown to be more effective and less toxic in nature, which could provide new sources of novel P-gp inhibitors [19,20,21]. The natural molecules most recently identified as potential inhibitors of P-gp are part of the fourth generation of inhibitors. These compounds such as flavonoids, alkaloids, terpenoids, and saponins [22] are able to perform a double mode of action on P-gp: they can interfere with its transcriptional expression and/or directly inhibit its function. These compounds can be substrates for P-gp and compete with anticancer agents to bind to the active site of the transporter (drug-binding pocket, DBP) and reduce drug efflux. Furthermore, they can modulate the ATPase activity of the pump through interaction with the nucleotide-binding pocket (NBP) and, finally, they can interact with membrane lipids and alter the membrane environment. Fluidity and lipid density are two characteristics of the plasma membrane that can interfere with MDR. It is known how some natural molecules can modulate the activity of the proteins incorporated in the membrane through global perturbations of the lipid environment [23]. This alteration in the membrane fluidity could also influence the correct interaction of different substrates with the efflux systems, such as P-gp. Phenolic phytochemicals are promiscuous modifiers of membrane protein functions, suggesting that some of their actions may be due to a common membrane bilayer-mediated mechanism. The incorporation of pentacyclic triterpenes into the phospholipid film significantly changes its morphology. They are structurally similar to steroids, therefore, analogously to cholesterol and plant sterols, they are easily included in biomembranes, although they exert a more fluidizing effect [24]. P-gp inhibitors of plant origin, belonging to various classes of secondary metabolites, were identified. Several published reviews summarize the most relevant advances in plant-focused P-gp inhibition research in different tumor models [22,25,26,27]. There are many examples of the ability of natural molecules, as fourth-generation inhibitors, to modulate P-gp in leukemia tumor models (Table 1). These studies provide examples of the potential use of natural substances in combination with chemotherapy, as a P-gp inhibitor, to increase the bioavailability of anticancer drugs in leukemia. While it is true that inhibition of the efflux transporter is essential to enhance the activity of natural compounds to reverse MDR, it is also true that non-specific inhibition can produce undesirable adverse effects on other essential cellular functions. Sometimes, inhibition of P-gp leads to excessive accumulation of cytotoxic drugs and decreased excretion rates, which in turn, produces toxicity to normal cell functions. Naturally occurring molecules have structural diversity, which provides a valuable tool in the search for highly targeted P-gp inhibitors. Many naturally occurring P-gp inhibitors have been observed to be very non-specific, although less toxic in nature. However, the use of natural molecules, instead of conventional synthetic molecules due to their structural diversity and non-specific binding to targets, can lead to undesirable pharmacokinetic changes, meaning extensive research is needed to establish the pharmacological characteristics of these molecules [28]. Indeed, there is no shortage of contradictory claims adding to the challenges of using natural products, such as quercetin, which, according to some studies, stimulates P-gp-mediated efflux and increases resistance to anticancer drugs in MDR cells [29,30]. Contrastingly, other studies have shown that quercetin is able to inhibit P-gp and decrease the resistance of anticancer drugs [31]; thus, it is necessary to evaluate all-natural molecules using the same standard methods, while all research needs to be more specific and focused to avoid such contradictions.

## 5. Four Natural Molecules Active in AML: Our Experience

This review will focus on the potential anticancer properties of specific compounds on which many research groups, including ours, have previously directed their studies: phytol, curcumin, lupeol, and heptacosane. In the present review, we have chosen to report data on four compounds that has exerted inhibitory effects on P-gp through various mechanisms of action. These compounds can inhibit the function of the efflux pump and/or inhibit its expression, yet they are all able to restore the sensitivity of the MDR cancer cells to chemotherapeutic drugs, exhibiting strong MDR reversal activity in an in vitro resistant AML model.

These four molecules have the common characteristic of being present in natural foods, such as phytol, which is in the chlorophylls of vegetables, curcumin is in curry, lupeol is in both vegetables and fruit, and heptacosane is in fruits; moreover, they can be assumed to be nutraceutical substances [50,51,52,53]. The list of therapeutic properties is certainly very long for curcumin and lupeol, whose anti-inflammatory, antioxidant, antimicrobial, antitumor, cardioprotective, antipyretic, analgesic, wound healing, anticonvulsant, antiarthritic, and antidiabetic activities are well known [54,55,56,57], whereas for phytol and heptacosane, the knowledge of those is limited. Furthermore, it is known that the dysfunction of the intestinal microbiome is associated with both the outcome of chemotherapy and adverse events, in many types of neoplastic tumors as well as in AML [58,59]. Many natural compounds are able to modify the response to therapy by modulating the microbiota [60,61,62]. It has been documented, for example, that there is a bidirectional regulation between the curcumin, which implements the intestinal bacterial flora, and itself which can biotransform the curcumin. Furthermore, in studies using AML animal models, curcumin seems to sensitize the response to Ara-C (cytarabine) by regulating the microbiota and controlling the level of cholesterol, which improves chemoresistance [63,64].

The choice of these four compounds, in particular, was made, above all, because they share a common target, whose prominent role in the resistance to antiblastic agents has been studied extensively by our group: NF-κB. Many natural substances, P-gp inhibitors, modulate its expression, through NF-κB, which includes P-gp as a target at the transcriptional level. Here, we present a few studies that have been performed in vitro on cell lines of different tumor types, which have been made resistant to the most common chemotherapeutic substrates of P-gp. S-Adenosylmethionine, a natural compound and a nutritional supplement, in addition to the numerous anticancer actions observed in different tumor models, is able to reverse multi-drug resistance, reducing the overexpression of P-gp by inhibiting the activation of NF-κB in colorectal cancer cell lines [65].

Among the natural polyphenols, the grape seed, procyanidin, seems to affect MDR mediated by P-gp overexpression, by acting on the NF-κB and MAPK/ERK pathways in ovarian cancer cell lines [66]. A similar result was obtained by Sun J. and colleagues who show that the natural compound clitocine is able to reverse P-gp-associated MDR via the downregulation of NF-κB in hepatoma and uterine cancer cell lines [67]. Feroniellin A, a furanocoumarin, reduces the expression of NF-κB and inhibits P-glycoprotein expression in lung cancer cell lines [68]. Moreover, in a breast cancer cell line, Zhu L. et al. demonstrate that Oroxylin A, a natural monoflavonoid, suppresses P-gp expression via the Chk2/P53/NF-κB signaling pathway [69].

With regards to the chosen natural substances, the mechanisms of action on NF-κB for both curcumin and lupeol are well known. However, phytol and heptacosane are major compounds in the essential oil of Euphorbia intisy, while in AML cells they have been shown previously to be able to reverse the MDR induced by P-gp, by acting both on its expression, through the inhibition of NF-κB, and on its efflux function [70].

## 6. Phytol

One of the most documented actions of phytol is its ability to interfere with the NF-κB pathway. This transcription factor has a preponderant role in all diseases characterized by a chronic inflammatory matrix, and its hyperactivation is found in many neoplastic diseases. Phytol has the ability to reduce inflammation by acting on important mediators, such as proinflammatory cytokines, and on the NF-κB pathway [71]. We have previously demonstrated a strong inhibition of NF-κB DNA-binding activity by phytol in a model of multi-drug resistant AML, characterized by overactivation of this factor [46]. Phytol is a precursor to produce synthetic forms of vitamin E and vitamin K1, and it is a component of many essential oils. There are several pieces of evidence that demonstrate how vitamins K1, K2, and K3 can potentially modulate important cellular responses in metabolic, inflammatory, and tumor diseases, such as free radical species of oxygen (ROS)-induced apoptosis, radical, and oxidative stress, alongside synthesis of metalloproteinases [72,73]. Similarly, essential oils with a high content of phytol have been shown to have therapeutic effects, for example, by showing antiproliferative effects on triple-negative breast cancer cell lines and on AML MDR cells [74,75,76,77,78,79]. In recent years, it has been proven that the phytol molecule is responsible for the numerous therapeutic effects observed [80,81]. In particular, a systematic review of the different properties of phytol underlines how this is characterized by anxiolytic, metabolism-modulating, cytotoxic, antioxidant, autophagy- and apoptosis-inducing, antinociceptive, anti-inflammatory, immune-modulating, and antimicrobial effects [51]. Although Bobe et al. indicates, in a systematic review, that 10 µM is the physiological concentration of phytol, and anything above this, phytol appears to cause toxicity in normal cells and mortality in animals [82]; however, our group did not observe toxicity in two normal-like cell lines, hTERT RPE-1 and 1-7HB2, until a concentration of 100 µg/mL (340 µM) [46]. Among the signal pathways, in which phytol intervenes, NF-κB is redundant. Labbozzetta et al. [46] show that in a study conducted in the AML MDR cell line, HL-60R, which is derived from HL-60 cells by selecting for doxorubicin resistance, phytol produces a strong decrease in the expression of P-gp at 25 µg/mL (84 µM), at both the mRNA and protein levels. This MDR cell line is characterized by overexpression of P-gp, hyperactivation of NF-κB, and pharmacological resistance to some antiblastic drugs, such as doxorubicin [83]. Phytol was able to reduce P-gp expression levels through inhibition of the NF-κB factor. Conversely, although molecular docking analysis showed the ability of phytol to interact with both sites of the P-gp drug-binding pocket (DBP) and the nucleotide-binding pocket (NBP), the doxorubicin accumulation analysis conducted in the HL-60R cell line, demonstrates how phytol does not seem to modulate the function of the efflux pump [46].

## 7. Curcumin

Curcumin is a phytochemical extracted from the dried rhizomes of Curcuma longa L. and is part of traditional herbal medicines. Much research has been done on curcumin. Its innumerable biological properties are widely known. Even if curcumin has the advantage of not being toxic (doses up to 3600–8000 mg daily for 4 months did not result in evident toxicities, except for mild nausea and diarrhea), unfortunately, it has the great disadvantage of not being very bioavailable after oral administration, with a systemic ineffectiveness. On the contrary, oral application can be useful for therapeutic action at the intestinal level, or when it is applied topically for local actions in tissues, such as skin and oral mucosa [84]. Mirzaei et al. summarize the results of several pharmacokinetic studies on curcumin. Here, it can be seen that oral administration determines the achievement of very low plasma concentrations, while intraperitoneal administration in mice and rats improves this aspect. Oral administration in humans, up to 10–12 mg/kg, determines the achievement of approximately 50/51.2 ng/mL [85]. The different formulations studied over the years have certainly led to an increase in the bioavailability of curcumin [55], as well as all vehicle systems into which curcumin has been incorporated. Tomeh et al. [86] examined all the delivery systems that have been analyzed in recent years to improve the stability and also cellular uptake of curcumin. Undoubtedly a system that transports the substance in a specific tissue or, in the case of liquid neoplasms such as leukemia, towards the tumor cells, could be a winning weapon. Many research groups, including ours, have analyzed many analogs with the aim of finding compounds that are equally or more effective, yet with higher in vivo stability [87,88,89]. Curcumin is endowed with different actions, which may explain its anti-inflammatory and antitumor properties [90]. In particular, it may interfere with both NF-κB activation and Akt kinase signaling. The pathway PI3K (phosphoinositide 3-kinases)/Akt/NF-κB seems to be the target of curcumin, from which its ability to inhibit the expression of P-gp derives [91]. The effects of curcumin on P-gp expression, both at the mRNA and protein levels, are documented in many MDR tumor models [34], although its action on the pump function is more interesting. The action of curcumin on P-gp function, in fact, was demonstrated by studies on rhodamine 123 (Rh123) accumulation and efflux in different tumor models, as well as retinoblastoma, breast cancer, human cervical carcinoma, and chronic myeloid leukemia cell lines [35,92,93,94]. In particular, Sreenivasan et al. [92] showed that curcumin-inhibited verapamil-stimulated ATPase activity of P-gp at higher concentrations and interacted at the substrate binding site of P-gp and not at the nucleotide-binding region. Analysis of the modulation of MDR by curcumin derivatives provided some evidence that confirms an inhibitory action on P-gp, both on its expression, mediated by three natural curcuminoids present in turmeric [95], and on its function, modulating P-gp mediated efflux through synthetic analogs of curcumin [35,96,97,98]. To date, no evidence has been reported on a possible modulation of the P-gp pump function by curcumin or its analogs in AML. In our preliminary study, conducted in an MDR AML cell line HL-60R and using a flow cytometry assay, we observed that curcumin causes an increase in the intracellular accumulation of doxorubicin, similar to the P-gp inhibitor verapamil. This led us to hypothesize that the modulating action of the function of P-gp occurs also in AML (data presented at the 41st National Congress of the Italian Society of Pharmacology, 16–19 November 2022).

## 8. Lupeol

AlQathama et al. [99] evaluated the antiproliferative and antimigratory abilities of 27 popular herbal infusions, in addition to the reverse P-gp efflux, in many types of tumor cell lines. Among the bioactive phytochemicals contained in these plants, which possess a better activity in terms of reversing MDR, lupeol is also present. Lupeol exhibits cytotoxic effects, particularly in AML cell lines [100,101]. These properties, together with the anti-inflammatory ones, seem to depend on its specific action on the NF-κB pathway, as shown in many studies concerning neoplastic and chronic inflammatory diseases [102,103,104].

Recently, Fontana et al. [44] confirmed this mechanism of action in two AML cell lines, one of which was responsive and the other exhibited MDR. They showed that in the MDR cell line, the strong inhibition of NF-κB activation led to a reduction in P-gp expression. However, the action of lupeol appears to be complex, involving multiple signaling pathways, which leads to the induction of apoptosis and cell growth inhibition.

Lupeol significantly reduces the expression of the Ras oncoprotein and modulates the protein expression of various signaling molecules involved in PKCα (protein kinase C alpha)/ODC (ornithine decarboxylase), PI3K/Akt and MAPKs (mitogen-activated protein kinase) pathways, in addition to strong reductions in the activation of the NF-κB signaling pathway [105,106,107] in some solid tumors. In analysis of a specific action of the P-gp function, in AML cell lines with an MDR phenotype, we observed a minimal increase in the accumulation of its substrate following pretreatment with lupeol. This result leads us to hypothesize that its action is also directed to the pump function, although with lower efficacy than previously observed in the P-gp expression (data presented on 41st National Congress of the Italian Society of Pharmacology, 16–19 November 2022).

Unfortunately, pharmacokinetic studies on lupeol in rats, mainly after oral administration, demonstrated a bioavailability of < 1% [108,109].

Different formulations able to enhance the solubility and oral bioavailability of lupeol, as well as more potent derivatives [45], are needed.

## 9. Heptacosane

There is little information regarding the biological properties of heptacosane, except those that report actions referable to phytocomplexes, for which it is one of the major constituents [110,111,112,113]. Heptacosane is one of the major components (8%) of the Centaurea baseri essential oil, Köse et al. have previously described its multiple activities, including its cytotoxic properties against MCF-7, PANC-1, A549, and C6 glioma cell lines [114]. Antitumor properties of the *Euphorbia intisy* essential oil on AML cells were likewise investigated, and the specific actions of its two major components phytol and heptacosane were evaluated [57]. In contrast to phytol, heptacosane does not modulate the NF-κB pathway but reduces the expression levels of some of the NF-κB targets. As suggested by a docking study on P-gp, heptacosane appears to have the most favorable interaction with the drug binding pocket site (DBP), whereby it assumes a conformation that could close the efflux pump. The compound also binds to the nucleotide-binding pocket (NBP) site, similar to verapamil, the classic reference drug used in P-gp functionality studies [115,116].

Heptacosane acts both as an ATP-dependent drug efflux transporter stimulator and as a substrate for P-gp transport. It was shown to stimulate the activity of the P-gp ATPase and concurrently inhibit the efflux function of the P-gp by acting in a competitive manner with another substrate [46].

## 10. Conclusions

First-, second-, and third-generation P-gp inhibitors have, for different reasons, proven unsuccessful in the clinic. Their main problem is owing to the high toxicity produced through the inhibition of the physiological functionality of P-gp in healthy tissues, in addition to the interference of these inhibitors with the metabolism of numerous substances. In this context, it could be interesting to intervene with low-toxic molecules able to modulate the overexpression of efflux pumps present in cells with the MDR phenotype. Another interesting strategy could be to use the delivery of efflux pump inhibitors, by administering nanoformulations to cells that overexpress them. This approach would allow us to avoid the undesirable effects on the physiological function of P-gp, providing natural inhibitors in nanosystems targeting cancer cells and resistant leukemic stem cells that overexpress efflux pumps, i.e., using this possibility in AML to target CD33 positive cells [117]. Here, it would be possible to specifically eradicate resistant clones.

Several recent papers indicate that natural substances such as phytol, curcumin, lupeol, and heptacosane, acting on P-gp expression and function, could be fully included among the fourth-generation inhibitors as they offer lower toxicity (Figure 1). These molecules deserve to be evaluated in vivo, as they could have implications in the clinic as adjuvants in chemotherapy regimens, especially in the niche category of patients with relapsing and resistant AML.

## Figures and Tables

**Figure 1 ijms-24-04140-f001:**
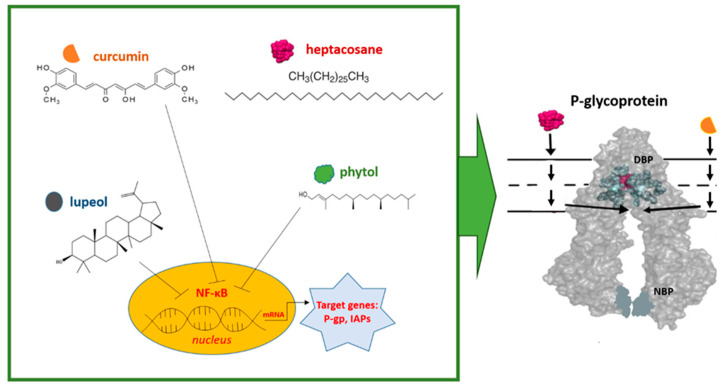
Graphical representation of the observed mechanisms of action of phytol, curcumin, lupeol, and heptacosane on the expression and function of P-gp in AML. Curcumin, phytol, and lupeol act by modulating the expression of P-gp through the inhibition of NF-κB. Curcumin is also able to modulate the function of P-gp through interaction with the DBP site. The modulation mechanism of heptacosane is instead linked to its interaction with the DBP and NBP sites (see text for explanation). (IAPs: apoptosis inhibitory proteins; DBP: drug-binding pocket; NBP: nucleotide-binding pocket).

**Table 1 ijms-24-04140-t001:** Some examples of natural inhibitors of P-glycoprotein in leukemia cell lines.

P-gp Natural Inhibitors “Fourth Generation”	Mechanism of Action
**Flavonoids**	
Quercetin	Inhibition of P-gp gene expression in adryamicin-resistant human chronic myeloid leukemia cell line (K562/ADR) [31].
Wogonin	Inhibition of P-gp expression in a human acute myeloid leukemia cell line (HL-60) [32].
Grape seed proanthocyanidin extract (GSPE)	GSPE inhibits P-gp expression via the PI3K / Akt signal transduction pathway in adryamicin-resistant human acute myeloid leukemia cell line (HL-60/ADR) [33].
Curcumin	Inhibition of P-gp mediated efflux in a doxorubicin-resistant human chronic myeloid leukemia cell line (K562/Dox) [34,35].
**Alkaloids**	
Pervilleine A	Inhibition of P-gp gene expression in a multidrug-resistant human T-cell childhood acute lymphocytic leukemia cell line (CEM/VLB_100_) [36].
Lobeline	Inhibition of P-gp mediated efflux probably by substrate competition in a multidrug-resistant human T-cell childhood acute lymphocytic leukemia cell line (CEM/ADR_5000_) [37].
harmine, and sanguinarine	Inhibition of P-gp gene expression in a multidrug-resistant human T-cell childhood acute lymphocytic leukemia cell line (CEM/ADR_5000_) [38].
Berbamine	Inhibition of mRNA and P-gp protein expression in an imatinib-resistant BCR-ABL-positive chronic myeloid leukemia cell line (K562-r) [39].
Reserpine and yohimbine	Inhibition of P-gp mediated efflux in a multidrug-resistant human T-cell childhood acute lymphocytic leukemia cell line (CEM/VLB_100_) [40].
Bromocriptine	Inhibition of the overexpressed P-gp protein in a vinblastine-resistant human chronic myeloid leukemia cell line (K562/R_10_) [41].
Isoquinoline alkaloid chelidonine	Inhibition of P-gp mediated efflux in multidrug-resistant human T-cell childhood acute lymphocytic leukemia cell line (CEM/DOX_5000_) [20].
**Terpenoids**	
Limonin	Inhibition of P-gp mediated efflux in a multidrug-resistant human T-cell childhood acute lymphocytic leukemia cell lines (CCRF-CEM and CEM/ADR_5000_) [42].
Euphodendroidin D and Pepluanin A	Inhibition of P-gp mediated efflux via binding with its active sites in a daunomycin-resistant human chronic myeloid leukemia cell line (K562/R_7_) [43].
Lupeol	Inhibition of mRNA and P-gp protein expression in a doxorubicin-resistant human acute myeloid leukemia cell line (HL-60R) [44,45].
Phytol and heptacosane	Inhibition of P-gp mediated efflux via binding with the DBP site, in a similar way to verapamil, in a doxorubicin-resistant human acute myeloid leukemia cell line (HL-60R) [46].
**Saponins**	
Ginsenoside F1	Inhibition of P-gp mediated efflux in daunorubicin- and doxorubicin-resistant acute myeloid leukemia sublines (AML-2/D_100_ and AML-2/DX_100_) [47].
Gracillin	Inhibition of P-gp mediated efflux via direct interaction with active binding sites in a daunorubicin-resistant human chronic myeloid leukemia cell line (K562/R_7_) [48].
Pinnatasterone	Inhibition of P-gp mediated efflux via binding with its active sites in a daunomycin-resistant human chronic myeloid leukemia cell line (K562/R7) [49].

CEM/VLB100 derived from CCRF-CEM cells with selection for vinblastine resistance; CEM/DOX5000 derived from CCRF-CEM cells with selection for doxorubicin resistance; CEM/ADR5000 derived from CCRF-CEM cells with selection for adriamycin resistance; AML-2/D100 and AML-2/DX100 derived from acute myeloid leukemia (OCI/ AML-2) cells with selection for daunorubicin and for doxorubicin resistances, respectively.

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
