# Peer review of "Natural Inhibitors of P-glycoprotein in Acute Myeloid Leukemia"

_ijms, 2023, doi:10.3390/ijms24044140_

Round 1
Author Response
"Please see the attachment."

Reviewer 2 Report
In this manuscript, Labbozzetta et al. examine the advantages of using natural substances as P-gp inhibitors (precisely - phytol, curcumin, lupeol and heptacosane) for AML therapy. These specific compounds were selected build on authors’ previous work with these compounds in AML cell lines.
`
I have some comments that should be addressed by the Authors:
In this work the Authors intend to review natural inhibitors and their effect on P-glycoprotein function in AML. However, as Authors state that “to date no evidence has been reported on a possible modulation of P-gp pump function by curcumin in AML” (lines 229-231). Therefore, it is unclear why curcumin was chosen for this review as data of the effect of this chemical on AML is very scarce. It seems as Authors wanted to mention their work and self-cite, which can not be encouraged. Furthermore, in section 6, where curcumin is discussed, a third of citations cite review articles but not original research, which also does not seem appropriate.
In general, choice of chemicals which are discussed in the context of AML is questionable as Authors indicate that “the choice of these four compounds in particular was made above all as they have a common target: NF-κB” (lines 152-153). It should be taken with the grain of criticism, as probably majority if not all cyto-active compounds affect NF-κB pathway. Accordingly, this work could bring more significant contribution to the field if the most effective natural P-gp inhibitors in AML were reviewed, not those compounds that have been studied by the Authors themselves. Furthermore, Authors should discuss P-gp inhibitors’ effect more specifically on AML as recent publication by Shah et al (2022) already thoroughly reviewed how natural compounds inhibit the resistance of chemotherapeutic drugs in general.
Shah, D., Ajazuddin, & Bhattacharya, S. (2022). Role of natural P-gp inhibitor in the effective delivery for chemotherapeutic agents. Journal of cancer research and clinical oncology, 10.1007/s00432-022-04387-2. Advance online publication. https://doi.org/10.1007/s00432-022-04387-2
P-gp distribution in normal human tissues and P-gp inhibitors’ effect on normal cells should also be discussed, at least briefly.
Paragraph (lines 148-151) about the intestinal microbiome and its modulation by natural compound must be expanded and specified in order to fit the scope of this review, otherwise this paragraph is redundant.
Limitations of natural P-gp inhibitors in cases of AML also should be more thoroughly discussed.
Figure 1 is low resolution, which should be improved. In addition, placement position of Figure 1 does not seem appropriate.
Minor revisions:
Some typos and grammar need to be corrected (lines 19 and 231 – periods missing, line 180 – “interfere with” not “interfere on”, ecc)).
The last paragraph in section 4 seems to be redundant.
Paragraphs consisting of one sentence should be avoided (lines 33-34, 47-50).
Overall, recommend reconsider after major revision.
Author Response
"Please see the attachment."

Round 2
Reviewer 1 Report
Article report 2, IJMS_2146275:
Major comment 1: The best part of the first version of the manuscript was the table summarizing information on natural inhibitors of P-glycoprotein and their effects in leukemia cell lines. As information on Phytol, curcumin, lupeol and hepatcosane was missing in the table, I had requested to add these in table I. However, there is no table in the revised version of the manuscript. To make things worse the table has been replaced by a muddled flood of words. This is unacceptable.
If there will be a second revision it should feature table I with the added information:
1. The docking study of the possible binding modes of phytol and heptacosane with the P-gp indicated that the two compounds bind the DBP site in a similar way to verapamil in Labbozzetta et al., Pharmaceuticals 2022, 15, 356.
2. Curcumin as a Modulator of P-Glycoprotein in Cancer: Challenges and Perspectives in LopesRodrigues et al., Pharmaceuticals (Basel). 2016 Nov 10;9(4):71.
3. Lupeol and its derivatives as anticancer and anti-inflammatory agents: Molecular mechanisms and therapeutic efficacy in Liu et al., Pharmacol Res. 2021 Feb; 164:105373.
Major comment 2: Chapter 2 is lacking information on P-gp in AML cells. This information has been given in the response to my first report, and should be added including the references.
Many types of cancer express ABC transporters, such as colorectal cancer, liver cancer, acute myeloid (AML) and breast cancer. In which approximately 35% AML the overexpression of P-gp causes the failure of anticancer therapies. P-g expression has been detected in approximately one-third of patients with AML at diagnosis and more than half of patients in remission; higher levels have been observed in certain subtypes, including secondary leukemia. In AML, P-gp expression is associated with a lower rate of complete remission, disease-free survival, and overall survival.
Amawi H, Sim HM, Tiwari AK, Ambudkar SV, Shukla S. ABC Transporter-Mediated Multidrug-Resistant Cancer. Adv Exp Med Biol. 2019; 1141:549-580. doi: 10.1007/978-981-13-7647-4_12. PMID: 31571174.
Vasconcelos FC, de Souza PS, Hancio T, de Faria FCC, Maia RC. Update on drug transporter proteins in acute myeloid leukemia: Pathological implication and clinical setting. Crit Rev Oncol Hematol. 2021 Apr; 160:103281. doi: 10.1016/j.critrevonc.2021.103281. Epub 2021 Mar 2. PMID: 33667660.
Gao F, Dong W, Yang W, Liu J, Zheng Z, Sun K. Expression of P-gp in acute myeloid leukemia and the reversal function of As2O3 on drug resistance. Oncol Lett. 2015 Jan;9(1):177-182. doi: 10.3892/ol.2014.2692. Epub 2014 Nov 10. PMID: 25435954; PMCID: PMC4247107.
Minor comments:
1. Considering their responses, it seems that the authors are lacking the ability to calculate drug concentrations which lessens the relevance of their results.
2. Facts: Phytol (MW 300g/Mol) 25 µg/mL = 83 mM, Heptacosane (MW 380 g/Mol) 50 µg/mL = 130mM.
3. The authors describe in vitro effects for phytol at 83 mM and heptacosane at 130mM. This is 10 times above the physiological range (<10 mM).
4. In their response the authors argue that other studies have used phytol at 10mM, 1000 times above the physiological range (<10 mM). This argument is invalid as in vitro effects at 10mM are even less relevant than at 100mM.
5. Lane 362. A system that vehicles a substance… vehicle is not a verb in English. The correct verb would be transport or deliver. A system that transports a substance toward the tumor.

Author Response
Major comment 1: The best part of the first version of the manuscript was the table summarizing information on natural inhibitors of P-glycoprotein and their effects in leukemia cell lines. As information on Phytol, curcumin, lupeol and hepatcosane was missing in the table, I had requested to add these in table I. However, there is no table in the revised version of the manuscript. To make things worse the table has been replaced by a muddled flood of words. This is unacceptable.
If there will be a second revision it should feature table I with the added information:
- The docking study of the possible binding modes of phytol and heptacosane with the P-gp indicated that the two compounds bind the DBP site in a similar way to verapamil in Labbozzetta et al., Pharmaceuticals 2022, 15, 356.
- Curcumin as a Modulator of P-Glycoprotein in Cancer: Challenges and Perspectives in LopesRodrigues et al., Pharmaceuticals (Basel). 2016 Nov 10;9(4):71.
- Lupeol and its derivatives as anticancer and anti-inflammatory agents: Molecular mechanisms and therapeutic efficacy in Liu et al., Pharmacol Res. 2021 Feb; 164:105373.
We thank the reviewer for his/her kind suggestion. We have reformulated the paragraph, reintroduced the table and added the references.
Major comment 2: Chapter 2 is lacking information on P-gp in AML cells. This information has been given in the response to my first report, and should be added including the references.
Many types of cancer express ABC transporters, such as colorectal cancer, liver cancer, acute myeloid (AML) and breast cancer. In which approximately 35% AML the overexpression of P-gp causes the failure of anticancer therapies. P-g expression has been detected in approximately one-third of patients with AML at diagnosis and more than half of patients in remission; higher levels have been observed in certain subtypes, including secondary leukemia. In AML, P-gp expression is associated with a lower rate of complete remission, disease-free survival, and overall survival.
Amawi H, Sim HM, Tiwari AK, Ambudkar SV, Shukla S. ABC Transporter-Mediated Multidrug-Resistant Cancer. Adv Exp Med Biol. 2019; 1141:549-580. doi: 10.1007/978-981-13-7647-4_12. PMID: 31571174.
Vasconcelos FC, de Souza PS, Hancio T, de Faria FCC, Maia RC. Update on drug transporter proteins in acute myeloid leukemia: Pathological implication and clinical setting. Crit Rev Oncol Hematol. 2021 Apr; 160:103281. doi: 10.1016/j.critrevonc.2021.103281. Epub 2021 Mar 2. PMID: 33667660.
Gao F, Dong W, Yang W, Liu J, Zheng Z, Sun K. Expression of P-gp in acute myeloid leukemia and the reversal function of As2O3 on drug resistance. Oncol Lett. 2015 Jan;9(1):177-182. doi: 10.3892/ol.2014.2692. Epub 2014 Nov 10. PMID: 25435954; PMCID: PMC4247107.
We thank the reviewer for his/her suggestion. We have added the informations in the text, including the references.
Minor comments:
- Considering their responses, it seems that the authors are lacking the ability to calculate drug concentrations which lessens the relevance of their results.
Sorry for the continued misunderstandings, we are sure of the correctness of our calculation.
- Facts: Phytol (MW 300g/Mol) 25 µg/mL = 83 mM, Heptacosane (MW 380 g/Mol) 50 µg/mL = 130mM.
We have reported the calculations of the concentrations of phytol and heptacosane carried out and reported by us in brackets in the text of the paper (lane 370 and 375).
- Phytol (MW 296.53) 25 µg/mL = 0.025 g/L
0.025/296.53= 8.4×10-5 M= 84 µM
100 µg/mL = 0.1 g/L
0.1/296.53= 0.34×10-3 M = 340 µM
- Heptacosane (MW 380.73) 50 µg/mL = 0.05 g/L
0.05/380.73 = 0.13 × 10-3 M = 130 µM
- The authors describe in vitro effects for phytol at 83 mM and heptacosane at 130mM. This is 10 times above the physiological range (<10 mM).
We thank the reviewer for his/her clarification. We reported in vitro effects for phytol at 84 µM as published in Labbozzetta et al.( Pharmaceuticals 2022, 15, 356), above the physiological range (<10 µM), but according with Bobe at al. (Eur J Cancer Prev. 2020 29:191-200), in which it is reported a range of cytotoxic effects at concentrations between 5 and 125 µM in different cancer cell lines.
- In their response the authors argue that other studies have used phytol at 10mM, 1000 times above the physiological range (<10 mM). This argument is invalid as in vitro effects at 10mM are even less relevant than at 100mM.
Sorry again, but the argument is relevant as we reiterate that the concentration of phytol used by us is 84 µM and not 84 mM.
- Lane 362. A system that vehicles a substance… vehicle is not a verb in English. The correct verb would be transport or deliver. A system that transports a substance toward the tumor.
We thank the reviewer for his/her suggestion. We have correct it in the text.
Reviewer 2 Report
The authors have revised the manuscript thoroughly and, in my opinion, it is now acceptable for publication.
Author Response
Thanks
Round 3
Reviewer 1 Report
Lane 264: After all, Shariare et al, show an IC50 of pure phytol on an AML cell line of about 10 mM [83]. This information is irrelevant and misleading, as 10mM phytol is 1000 times above the physiological range. The sentence and the reference have to be removed.
Author Response
We have removed the sentence and reference 83.